# Plasma Concentrations of Neurofilament Light Chain Protein and Brain-Derived Neurotrophic Factor as Consistent Biomarkers of Cognitive Impairment in Alcohol Use Disorder

**DOI:** 10.3390/ijms24021183

**Published:** 2023-01-07

**Authors:** Nerea Requena-Ocaña, Pedro Araos, Pedro J. Serrano-Castro, María Flores-López, Nuria García-Marchena, Begoña Oliver-Martos, Juan Jesús Ruiz, Ana Gavito, Francisco Javier Pavón, Antonia Serrano, Fermín Mayoral, Juan Suarez, Fernando Rodríguez de Fonseca

**Affiliations:** 1Neuropsychopharmacology Group, Instituto IBIMA, Hospital Regional Universitario de Málaga, Avda. Carlos Haya 82, Sótano, 29010 Malaga, Spain; 2School of Psychology, Complutense University of Madrid, Campus de Somosaguas, 28040 Madrid, Spain; 3Andalusian Network for Clinical and Translational Research in Neurology (NEURO-RECA), 29010 Malaga, Spain; 4Neurology Service, Regional University Hospital of Malaga, 29010 Malaga, Spain; 5Institute D, Research in Health Sciences Germans Trias i Pujol (IGTP), Addictions Unit-Internal Medicine Service, Campus Can Ruti, Carrer del Canyet s/n, 08916 Badalona, Spain; 6Provincial Drug Addiction Center (CPD) of Malaga, Provincial Council of Malaga, C/Ana Solo de Zaldívar, n3, 29010 Malaga, Spain; 7Center for Biomedical Research in the Cardiovascular Diseases Network (CIBERCV), Carlos III Health Institute, Calle de Melchor Fernández Almagro, 3, 28029 Madrid, Spain; 8Mental Health Clinical Management Unit, Institute of Biomedical Research of Malaga-IBIMA, Regional University Hospital of Málaga, 29010 Malaga, Spain; 9Department of Anatomy, Legal Medicine and History of Science, School of Medicine, University of Malaga, Boulevard Louis Pasteur 32, 29071 Malaga, Spain

**Keywords:** alcohol use disorder, addiction, neurodegeneration, cognitive impairment, dementia, neurofilament light chain protein, brain-derived neurotrophic factor

## Abstract

For a long time, Substance Use Disorders (SUDs) were not considered a component in the etiology of dementia. The fifth edition of the Diagnostic and Statistical Manual of Mental Disorders introduced substance-induced neurocognitive disorders, incorporating this notion to clinical practice. However, detection and monitoring of neurodegenerative processes in SUD patients remain a major clinical challenge, especially when early diagnosis is required. In the present study, we aimed to investigate new potential biomarkers of neurodegeneration that could predict cognitive impairment in SUD patients: the circulating concentrations of Neurofilament Light chain protein (NfL) and Brain-Derived Neurotrophic Factor (BDNF). Sixty SUD patients were compared with twenty-seven dementia patients and forty healthy controls. SUD patients were recruited and assessed using the Psychiatric Research Interview for Substance and Mental (PRISM) and a battery of neuropsychological tests, including the Montreal Cognitive Assessment test for evaluation of cognitive impairment. When compared to healthy control subjects, SUD patients showed increases in plasma NfL concentrations and NfL/BDNF ratio, as well as reduced plasma BDNF levels. These changes were remarkable in SUD patients with moderate–severe cognitive impairment, being comparable to those observed in dementia patients. NfL concentrations correlated with executive function and memory cognition in SUD patients. The parameters “age”, “NfL/BDNF ratio”, “first time alcohol use”, “age of onset of alcohol use disorder”, and “length of alcohol use disorder diagnosis” were able to stratify our SUD sample into patients with cognitive impairment from those without cognitive dysfunction with great specificity and sensibility. In conclusion, we propose the combined use of NfL and BDNF (NfL/BDNF ratio) to monitor substance-induced neurocognitive disorder.

## 1. Introduction

Dementia is considered a public health priority because of its alarming increase in prevalence (7.7 million/year), huge economic impact, and overwhelming consequences for family and caregivers. It is estimated that dementia will affect 65.7 million people by 2030 and 115.4 million by 2050 [1]. In this regard, lifetime substance abuse has been associated with progressive cognitive decline and neuronal injury across the lifespan [2,3]. Although Substance Use Disorders (SUDs) have not been considered an essential component in the etiology of dementia for a long time, the fifth edition of the Diagnostic and Statistical Manual of Mental Disorders (DSM-5) [4] introduced major neurocognitive disorder as equivalent to the concept of dementia, with a subtype related to substance or medication use [5]. In this regard, an Australian study reported that the most common clinical diagnosis for early onset dementia was alcohol-related dementia (18.4%), followed by Alzheimer’s disease (17.7%), vascular dementia (12.8%), and frontotemporal dementia (11.3%), which were more frequent in people aged between 45 and 64 years [6]. Furthermore, a prospective study has reported that Alcohol Use Disorder (AUD) is the strongest risk factor for the early onset of dementia and could be a contributing factor to all known dementias in men and women [7]. However, the mechanistic of this dementia-promoting action of alcohol is complex, if we consider the multi-morbidity associated with alcohol abuse. As an example, AUD generates an estimated annual mortality of 3 million people (5.3% of deaths) [8], which are associated with death from epilepsy (53%), hemorrhagic stroke (48%), and ischemic stroke (19%) [9]. All these disorders, together with malnutrition, psychiatric comorbidity, and neuroinflammation, can contribute to AUD-associated dementia.

Patients with AUD have several neuropsychological alterations related to processing speed, sustained attention, learning and memory, and global executive functions (including disinhibition, cognitive flexibility, planification, resolution strategies, and working memory) [10,11,12]. However, as stated above, the identification of alcohol-related cognitive impairment is a complex challenge because of its multifactorial etiologies. Among AUD patients, there is a high prevalence of Alzheimer’s disease (recognized by tau, phosphorylated tau, and amyloid β) [13] and Wernicke–Korsakoff syndrome (associated with thiamine) [14,15], but in the absence of these diagnoses, cognitive dysfunction may be present [16]. The explanation of why these biomarkers are insufficient in this population could be due to the influence of additional factors in cognitive dysfunction, such as other nutritional deficiencies (i.e., low BMI, ascorbic acid deficiency, or thiamine deficiency) [17,18], comorbid psychiatric disorders (i.e., affective disorders) [19], as well as inflammation and oxidative stress caused by chronic alcohol intake or alcohol withdrawal episodes per se [20]. Heavy alcohol consumption throughout life triggers a proinflammatory organic state that leads to neurocognitive alterations [21,22,23]. Alcohol can stimulate Toll-like receptor 4 (TLR4), which activates several signaling pathways (i.e., Nuclear Factor-κB, inducible Nitric Oxide Synthase), resulting in the release of cytokines, chemokines, and oxidative–nitrosative stress [24,25,26], associated with neuroinflammation and structural brain damage [27,28,29].

On the other hand, the background on cognitive impairment and dementia-related processes in cocaine and cannabis use disorders is still scarce and inconsistent [30,31]. Although some studies suggest that prolonged cocaine use is related to deterioration in some cognitive domains, these could be attributable to concomitant alcohol use [32]. However, patients with cocaine abuse manifest severe cocaine-induced cardiovascular consequences such as vasoconstriction, endothelial dysfunction, arteriosclerosis [33], and increases in oxidative stress [34] that could underlie cognitive dysfunction [35,36]. Moreover, acute cocaine use could also be a contributing factor to stroke risk in young people [37]. Similarly, although heavy cannabis use appears to lead to neuropsychological impairment, these associations might be attenuated or not significant when confounding variables are controlled (i.e., other substance use, psychiatric disorders, or psychosocial variables) [38].

Clinical studies have recognized Neurofilament Light chain protein (NfL), a structural component of the axonal cytoskeleton, as a promising biomarker for several neurodegenerative diseases, including mild cognitive impairment, Alzheimer’s disease, frontotemporal dementia, Parkinson’s disease, and amyotrophic lateral sclerosis [39,40]. NfL evaluation has provided superior capabilities to discriminate neurodegenerative disease in younger individuals (<65 years) in the absence of psychiatric influence [39]. Therefore, NfL levels could be a promising biomarker, since we still lack robust biological parameters to diagnose substance-induced neurocognitive disorders. Thus, a recent study reported elevated NfL levels in abstinent AUD patients, which were associated with cognitive impairment and the extent of white-matter lesions based on magnetic resonance imaging analysis [41]. In addition, Brain-Derived Neurotrophic factor (BDNF), a member of the neurotrophin family of growth factors, participates in the maintenance and growth of neurons through synaptogenesis, neurogenesis, gliogenesis, and axodendritic arborization, among others [42]. Thus, we have found that monitoring BDNF could help to identify AUD patients with and without cognitive impairment with high accuracy [43,44]. However, BDNF has been proposed as a broad biomarker of several neuropsychiatric disorders that are highly comorbid with AUD and/or cognitive impairment, especially depression [45], posttraumatic stress disorder [46], schizophrenia [47], cocaine use disorder [48,49], Parkinson’s disease [50] and Alzheimer’s disease [51].

The aim of this descriptive clinical study was to establish robust biomarkers to detect substance-induced neurocognitive disorders through neurodegenerative and neuro-regenerative factors (NfL and BDNF, respectively) which could be useful at a preventive level.

## 2. Results

### 2.1. Sociodemographic Characteristics and Clinical Variables of SUD, Dementia and Control Groups

A baseline socio-demographic description of the study participants is summarized in Table 1. Due to the huge differences between the SUD group and the dementia group, we were unable to match the three groups for age (*p* < 0.001) and sex (*p* = 0.001), except for BMI. However, the control group was selected to be as similar as possible between the SUD and dementia groups. Significant differences were observed between the three sample groups with respect to educational level (*p* = 0.002) and occupation (*p* < 0.001).

### 2.2. Clinical Characteristic of the Abstinent SUD Patients

Clinical characteristics of the SUD group are described in Table 2. Among patients attending outpatient treatment for substance abuse, AUD was the most prevalent (78%), followed by cocaine (60%) and cannabis use disorders (41%). There was a high prevalence of other comorbid SUD (60%) in patients with AUD, especially cocaine (48%) and cannabis (27%) use disorders. There was a high prevalence of other comorbid psychiatric disorders (76.7%), with lifetime mood and anxiety disorders being the most frequent, diagnosed in 52% and 35% of AUD patients, respectively. Furthermore, 85% of the abstinent SUD patients received psychiatric medication during the last year, most frequently being prescribed antidepressants (43%), anxiolytics (57%), and disulfiram (18%). Neuropsychological evaluation revealed that 56.7% of the SUD group showed some deficits related to global cognition (assessed by MoCA). Specifically, 27% seemed to have mild cognitive impairment, 22% had moderate cognitive impairment, and 8% had severe cognitive impairment.

### 2.3. Plasma Concentrations of NfL and BDNF in the SUD, Dementia and Control Groups

The impact of substance abuse on plasma concentrations of NfL, BDNF and NfL/BDNF ratio was studied in the total sample using a one-way ANCOVA with “group” (SUD, dementia, and control groups) as a factor and age as a covariate. Plasma concentrations of NfL, BDNF, and NfL/BDNF ratio were significantly different between groups (F_2,123_ = 16.409, *p* < 0.001, ηp^2^ = 0.211, F_2,97_ = 16.264, *p* < 0.001, ηp^2^ = 0.251, F_2,97_ = 22.326, *p* < 0.001, ηp^2^ = 0.315, respectively). Plasma concentrations of NfL were significantly increased in the SUD group (*p* = 0.001) and the dementia group (*p* < 0.001) compared to the control group (Figure 1A). Plasma concentrations of BDNF were decreased in the SUD group (*p* < 0.001) compared to the control group (Figure 1B). The NfL/BDNF ratio was significantly increased in the SUD group (*p* < 0.001) and the dementia group (*p* = 0.010) compared to the control group (Figure 1C).

### 2.4. Plasma Concentration of NfL and BDNF in Abstinent SUD Patients with and without Cognitive Impairment Compared to Dementia and Control Groups

We wanted to investigate how cognitive integrity (assessed by MoCA) in SUD patients (SUD + CI vs. SUD − CI) could affect plasma concentrations of NfL, BDNF, and the NfL/BDNF ratio using a one-way ANCOVA with “group” (SUD + CI, SUD − CI, dementia and control groups) as a factor and age as a covariate. When we analyzed the SUD + CI group, we found that plasma concentrations of NfL, BDNF, and NfL/BDNF ratio were significantly different between groups (F_3,122_ = 12.499, *p* < 0.001, ηp^2^ = 0.235; F_2,96_ = 11.821, *p* < 0.001, ηp^2^ = 0.270; F_3,96_ = 16.154, *p* < 0.001, ηp^2^ = 0.335, respectively). We observed higher NfL levels in the SUD + CI (*p* < 0.001) and the dementia group (*p* = 0.001) compared to the control group (Figure 2A). We also found lower BDNF levels in the SUD + CI (*p* < 0.001) and in the SUD − CI group (*p* = 0.003) compared to the control group (Figure 2B). The NfL/BDNF ratio was increased in the SUD + CI group (*p* < 0.001) and in the SUD − CI (*p* = 0.001) compared to the control group (Figure 2C).

### 2.5. Plasma Concentration of NfL and BDNF According to the Severity of Cognitive Impairment in Abstinent SUD Patients

We also measured cognitive functioning (assessed by MoCA) following chronic substance consumption in plasma concentrations of NfL and BDNF in the SUD group using a one-way ANCOVA with “cognitive impairment” (non-cognitive impairment, mild cognitive impairment, and moderate/severe cognitive impairment) as a factor and age as a covariate. Plasma concentrations of NfL, BDNF, and NfL/BDNF ratio were significantly affected by the severity of cognitive impairment (F_2,56_ = 3.545, *p* = 0.036, ηp^2^ = 0.112; F_2,34_ = 7.284, *p* = 0.002, ηp^2^ = 0.294; F_2,35_ = 6.108, *p* = 0.005, ηp^2^ = 0.259, respectively). There were significantly higher plasma concentrations of NfL (*p* = 0.032, Figure 3A) and lower levels of BDNF (*p* = 0.002, Figure 3B) in SUD patients with moderate–severe cognitive impairment than in those without cognitive impairment. NfL/BDNF ratio was higher in SUD patients with moderate/severe cognitive impairment compared with AUD patients without cognitive impairment (*p* = 0.012) or mild cognitive impairment (*p* = 0.018, Figure 3C).

Furthermore, we investigated if comorbid anxiety disorder could influence the effect of cognitive impairment on NfL levels using a one-way ANCOVA with “cognitive impairment” and “comorbid anxiety disorder” as factors and age as a covariate. We found significant effects of “cognitive impairment” and “comorbid anxiety disorder” in the NfL levels of SUD patients (F_2,53_ = 4.097, *p* = 0.022, ηp^2^ = 0.134; F_2,53_ = 4.944, *p* = 0.030, ηp^2^ = 0.085, respectively) but not for their interaction (F_2,53_ = 0.663, *p* = 0.520, ηp^2^ = 0.024). Thus, even controlling for comorbid anxiety disorder, the effect of cognitive impairment on NfL concentrations in SUD patients remains significant.

### 2.6. Correlation Analysis of NfL and BDNF according to Addiction-Related Variables in the SUD Group

Correlation analyses between addiction-related variables and plasma concentrations of NfL and BDNF were performed to explore the effect of specific substances on these parameters using partial correlations controlling for age (rho) (Appendix A). Interestingly, we found significant correlations between NfL concentrations and age at first alcohol use (rho = 0.315, *p* = 0.031), age at onset of AUD (rho = 0.455, *p* = 0.001) and length of AUD diagnosis (rho = 0.375, *p* = 0.010). We also showed significant correlations between NfL concentrations and age at first cocaine use (rho = 0.437, *p* = 0.008) and with the severity of sedative use disorder (rho = −0.815, *p* = 0.014). In addition, we found a positive and significant correlation between BDNF levels and the severity of cannabis use disorder (rho = 0.605, *p* = 0.017). Correlation analysis between plasma concentrations of NfL and BDNF according to addiction-related variables in the SUD group can be found in Appendix A.

### 2.7. Predictive Variables of Cognitive Impairment in SUD Patients

We generated a binary logistic regression model to discriminate between SUD patients with and without cognitive impairment (assessed by MoCA). In the final model, the variables included in the first step were “age”, “NfL/BDNF ratio”, “age at first alcohol use”, “age at onset of AUD”, “length of AUD diagnosis”, “age at first cocaine use”, “sedatives severity criteria”, and “cannabis severity criteria”. The model was prepared using the backward stepwise method and the predictive covariates were restricted to five, which were “age”, “NfL/BDNF ratio”, “age at first alcohol use”, “age at onset of AUD”, and “length of AUD diagnosis”. The Hosmer–Lemeshow test indicated good calibration (X^2^ = 5.905; *p* = 0.658) and was able to explain the variation of the dependent variable in 69.7% of the cases according to the Nagelkerke R2 method. It had a classification percentage of 86.8% showing a high sensitivity to classify SUD patients with cognitive impairment (87%) and without cognitive impairment (86.7%). ROC curve analysis (AUC = 0.930) indicated a high discrimination power (Figure 4A). The scatter plot of the predictive probabilities for SUD patients indicated that the means were significantly different between both groups (U = 24, *p* < 0.001) (Figure 4B).

Binary logistic regression analysis was then used to assess the potential for the NfL/BDNF ratio alone to be a good predictor for discriminating between SUD patients with and without cognitive impairment (assessed by MoCA). The variables included in the first step were “age” and “NfL/BDNF ratio”. The model was performed using the backward stepwise method and the predictive covariates were restricted to both variables. The Hosmer–Lemeshow test indicated good calibration (X^2^ = 11.114; *p* = 0.195) and was able to explain the variation of the dependent variable in 58.2% of the cases, according to the Nagelkerke R2 method. It had a classification percentage of 79.5% showing a high sensitivity to classify SUD patients with cognitive impairment (79.2%) and without cognitive impairment (80%). ROC curve analysis (AUC = 0.897) indicated a high discrimination power (Figure 4C). The scatter plot of the predictive probabilities for SUD patients indicated that the means were significantly different between both groups (U = 37, *p* < 0.001) (Figure 4D).

### 2.8. Correlation Analysis of NfL and BDNF with Neuropsychological Tests in the SUD Group

Psychometric data obtained by the SUD group in each neuropsychological test are summarized in Table 2. The Z-scores of the immediate free recall in list A was −1.30 ± 1.0 and in the immediate free recall in list B was −1.09 ± 0.631, showing mild deficits in verbal learning (VLTSC). In addition, the Z-scores of short delay free recall, short delay cued recall, long delay free recall, and long delay cued recall were −1.39 ± 1.200, −1.30 ± 0.984, −1.03 ± 1.045, −1.18 ± 1.310, which demonstrated mild alterations in short- and long-term verbal memory that could not be prevented by semantic cues (VLTSC). The Z-score of the Digit span subtest was −1.18 ± 1.468, suggesting mild impairment in the phonological loop. The Z-score of the TMT B was 3.14 ± 2.74, indicating severe deficits related to impairment in executive attention and mental flexibility.

Moreover, as shown in Table 3, there were significant correlations between NfL and variables associated with interference [immediate free recall list B (VLTSC)], planning [serial strategies in immediate recall (VLTSC) and serial strategies in short-delay free recall (VLTSC)] and slightly with verbal short-term memory [short delay free recall (VLTSC)] (rho = −0.363, *p* = 0.038; rho = −0.374, *p* = 0.032; rho = −0.464, *p* = 0.007, rho = −0.316, *p* = 0.073, respectively). We observed a significant correlation between NfL and memory intrusions [intrusions in free recall, intrusions in cued recall (VLTSC)] (rho = 0.408, *p* = 0.018; rho = 0.362, *p* = 0.038, respectively). However, we found no correlations between BDNF and scores of cognitive assessments.

## 3. Discussion

Detection and monitoring of neurodegenerative processes in patients with chronic substance abuse throughout their lifetime remains a major clinical challenge. Given that cognitive dysfunction can predict treatment outcomes [52,53] and could eventually lead to some type of dementia in SUD patients [54], it is necessary to establish a standardized early screening in health centers to stratify at-risk individuals.

The present study supports that substance-induced neurocognitive disorder is not only associated with deficits in plastic/trophic factors (BDNF), but also with real structural brain damage (NfL). It is important to note that we had to control for confounding variables that could alter plasma concentrations of NfL and BDNF, such as psychiatric conditions (mood and anxiety disorders), psychotropic medication (antidepressants and anxiolytics), early withdrawal effects (days of abstinence), and age (years). NfL evaluation can discriminate between depression from neurodegenerative disorders. Our results also indicated that anxiety disorders can affect plasma concentrations of NfL in our SUD sample [39]. Similarly, some studies have reported that depressive and anxiety symptoms could worsen the NfL profile in neurodegenerative processes such as multiple sclerosis and Parkinson’s disease [55,56]. However, when we controlled for anxiety disorders in the analysis, the association between cognitive impairment and NfL concentrations remained significant in SUD patients. Furthermore, we observed a positive and significant correlation between NfL and age of participants. NfL concentrations are not only a strong biomarker for neurodegeneration but are also closely related to senescence [40]. Except for anxiety disorders and age, there were no other variables affecting levels of NfL, BDNF, or NfL/BDNF ratio in our sample.

NfLs are intermediate filaments (~10 nm in diameter), which lie between actin (6 nm in diameter) and myosin (15 nm in diameter) filaments. NfLs are an important constituent of the cytoskeleton scaffolding of the brain in both the central nervous system and peripheral neurons. NfLs are composed of four different polymers: (1) light chain (NfL, ≈70 kDa), (2) medium chain (NfLM, ≈150 kDa), (3) heavy chain NLs (NfLH, ≈200 kDa), and (4) an α-internexin in the central nervous system (≈66 kDa) and a peripherin in the peripheral nervous system (≈57 kDa). All these subunits share a common structure composed of an amino-terminal globular head domain, a central α-helical rod domain, and a carboxy-terminal tail. However, NfM and NfH also have strongly phosphorylated serine–proline–lysine repeats in the carboxy-terminal tail [40,57]. NfL is an indicator of neuroaxonal damage, which is the pathological substrate of permanent disability in several, if not all, acute and chronic neurological disorders [57]. When an axon is damaged, NfLs are released into the extracellular space, passing into the cerebrospinal fluid and into the blood in lower concentrations [58]. Therefore, NfLs seem to be a very promising candidate biomarker to reflect brain damage of all neurological diseases. Furthermore, NfLs could allow the monitoring of the neurocognitive disorders, as well as the effects of rehabilitation and pharmacological treatments [40,59]. On the other hand, NfL levels may not be as specific when it comes to distinguishing cognitive disorders in Alzheimer’s disease, such as β-amyloid and Tau [60]. However, NfL evaluation has shown to be useful in differentiating certain neurological disorders that might be misdiagnosed [i.e., stratifying atypical Parkinsonian disorder from Parkinson’s disease (86–95%)] [61]. In addition, NfL concentrations are useful for differentiating neurodegenerative diseases from psychiatric disorders [i.e., distinguishing frontotemporal dementia from depression (98%)] [62].

Our results indicated higher plasma concentrations of NfL in the SUD and dementia patients compared to controls. NfL changes were observed in SUD patients with cognitive impairment, especially in those with moderate/severe cognitive dysfunction. In addition, patients with a lifetime of chronic substance abuse did not differ from dementia patients in terms of their NfL profile. Thus, our study supports that substance-induced neurocognitive disorder might be neurodegenerative and dementia-like. Furthermore, NfL concentrations were correlated with addiction-related variables, especially those associated with AUD (age at first-time alcohol use, age at AUD diagnosis, and length of AUD diagnosis). Both patients who initiated alcohol use and developed AUD late in life and patients with prolonged AUD throughout their life had higher levels of NfL. These results may suggest two things: (1) starting to abuse alcohol later in life is a risk factor for neurodegeneration; (2) the longer AUD lasts, the more neurodegeneration it produces. This is consistent with clinical and preclinical studies in which authors have found higher concentrations of NfL in AUD [41,63]. In addition, a significant negative relationship has been found between gray matter thickness and circulating NfL in heavy alcohol use [64]. Similarly, a longitudinal twin study found that moderate/elevated alcohol consumption increased the risk of dementia by 57% compared to the twin with light consumption. Furthermore, heavy alcohol consumption was associated with the development of dementia 5 years earlier [65]. However, the risk of dementia derived from alcohol consumption could be underestimated and biased due to the high rates of death that are not considered in the studies [66]. Thus, the prevalence of alcohol-related dementia is not homogeneous, ranging from 8.70 per 100,000 to 25.6% [67].

On the other hand, growth factors are a set of proteins that play a crucial role in cell growth, proliferation, and differentiation in the central nervous system [68]. Particularly, BDNF interacts with the TrKB receptor, which belongs to the family of neurotrophin receptor tyrosine kinases. Its binding activates neuroprotective signaling cascades such as the inositol phosphatidic 3-kinase PI3K/Akt pathway, the mitogen-activated protein kinase (MAPK) pathway, the phospholipase C-gamma (PLC-gamma) pathway, and the guanosine triphosphate hydrolase (GTP-ase) pathway [42]. Thus, functions attributed to BDNF include regulation of neurogenesis, synaptogenesis, and gliogenesis. BDNF also controls long-term potentiation (LTP), which is a mechanism in the hippocampus that results in improved memory and cognition [69]. Our results suggested decreased plasma levels of BDNF in SUD patients compared to healthy controls. BDNF alterations were observed in both SUD patients with and without cognitive impairment, but especially in those with moderate/severe cognitive dysfunction. This might indicate that BDNF could be a biomarker of both substance abuse and substance-related cognitive impairment, as our group has reported in previous studies [43,44]. Interestingly, we did not observe differences in BDNF concentrations in dementia patients compared to controls. Thus, these results could suggest that disrupted trophic signaling could underlie the neuronal death and cognitive dysfunction observed in our SUD patients. In contrast, the neuroprotection of BDNF might not be enough to counteract brain damage in patients with dementia.

Therefore, we proposed the NfL/BDNF ratio as a new index for cognitive impairment that could have clinical significance in SUD patients. It combines the effect of a progressive loss of BDNF trophic action with the progressive neuronal loss associated with NfL release. Both parameters would help to establish an objective biomarker of neurocognitive disease. Our results showed that the NfL/BDNF ratio was higher in SUD and dementia patients compared to controls. Increased NfL/BDNF ratio were observed in both SUD patients with and without cognitive impairment, but in those with moderate/severe cognitive impairment, this index was notably higher. In other words, substance abuse throughout life (particularly alcohol) is not only toxic but also anti-regenerative. These results might indicate that BDNF could be involved in a compensatory mechanism against neuronal damage, but not in advanced stages, when neurocognitive disability is established. In this regard, an emerging line supports the role of BDNF/TrkB signaling pathway as a compensatory response that delays symptoms in the early stage of Alzheimer’s disease, while in later stages, they would not be sufficient to prevent neurodegeneration [70,71,72]. In addition, we tested the NfL/BDNF ratio as a potential biomarker for substance-induced neurocognitive disorder in two binary logistic regressions. In the first model, “age”, “NfL/BDNF ratio”, “age at first alcohol use”, “age at onset of AUD”, and “length of AUD diagnosis” were variables able to stratify our SUD sample in cognitively impaired and non-cognitively impaired patients by 93%. In the second model, “age” and “NfL/BDNF ratio” were variables sufficiently capable of classifying our SUD sample in cognitively impaired and non-cognitively impaired patients by 89%.

Finally, 56.7% percent of our SUD patients had some degree of cognitive impairment, as similar studies have reported (31–50%) [73,74]. They had mild alterations in verbal learning, verbal short- and long-term memory, phonological loop, and severe deficits in executive attention and mental flexibility, as reported in other investigations [10,75]. These neuropsychological alterations have also been related to greater difficulties in maintaining abstinence and moderate alcohol consumption [76,77]. Furthermore, we found negative correlations between circulating NfL levels and interference, planning and verbal short-term memory, and a positive relationship with memory intrusions. Thus, we have demonstrated that neuronal structural damage underlies cognitive deficits induced by lifetime substance abuse, especially for alcohol abuse. This is in line with a recent meta-analysis that has revealed gray matter degeneration in the right cingulate gyrus, left middle frontal gyrus, and right insula in alcohol-dependent patients [78]. Lifetime alcohol abuse has also been consistently associated with white-matter reduction in several regions involving frontal connections [79,80].

We conclude that the NfL, BDNF, and NfL/BDNF ratio might serve as promising plasma biomarkers for substance-induced neurocognitive disorder, especially in patients with chronic alcohol consumption throughout their lifetime. Nevertheless, this study has some limitations that should be considered in future research. First, a large sample size and a prospective design are needed to test and establish the NfL, BDNF, and NfL/BDNF ratio as potential diagnostic biomarkers of neurodegeneration in the SUD population. Second, a stratification of the SUD group is necessary according to the type of drug use (alcohol, cocaine, cannabis, etc.). Finally, the study lacks significant representation of the female population, which precludes investigation of gender/sex differences in NfL and BDNF markers.

## 4. Materials and Methods

### 4.1. Recruitment and Screening of Participants

The present study included 60 abstinent SUD patients (SUD group) in outpatient treatment, 27 patients from neurology outpatient settings (dementia group), and 40 healthy control subjects (control group). SUD patients were recruited at the Centro Provincial de Drogodependencias (Málaga, Spain). Dementia patients were collected at the Neurology Service of the Hospital Regional Universitario de Málaga (Málaga, Spain). Control participants were included from databases of healthy subjects of the DNA National Biobank (Valencia, Spain).

To be included in the present study, the SUD group had to meet the following inclusion criteria: people ≥ 18 years in the abstinence phase (>1 month) with a diagnosis of SUD (alcohol, cocaine, cannabis, sedatives, and opioids). Exclusion criteria included personal history of long-term inflammatory disease or cancer, severe language limitations, pregnant or breast-feeding women, and infectious diseases such as HC, HB, and HIV. The dementia group was used as a clinical control of our SUD sample, as well as the healthy controls, because their high plasma NfL levels are well known. Thus, the dementia group had to meet the following inclusion criteria: people ≥ 60 years under neurologic treatment with a Mini-Mental State Examination (MMSE) score < 24 and willingness to participate by signing the informed consent. Exclusion criteria included personal history of alcohol use during the last year with a score > 8 on the Alcohol Use Disorder Identification Test (AUDIT), severe language limitations, and infectious diseases such as HC, HB, and HIV. Regarding the control group, participants with a history of substance abuse, comorbid psychiatric disorders, medical illness, and cognitive impairment were also excluded.

### 4.2. Ethical Statement

Informed consent was obtained from all subjects involved in the study. The ethical aspects of the core project (Proteomics of Cocaine Addiction: Central and Peripheral Biomarkers of Addiction) were approved by the Ethics and Clinical Research Committee of the Regional University Hospital of Malaga in accordance with the Ethical Principles for Medical Research with Human Subjects adopted in the Declaration of Helsinki by the World Medical Association (64th General Assembly of the WMA, Fortaleza, Brazil, October 2013), and Recommendation No. R (97) 5 of the Committee of Ministers to the Member States on the protection of medical data (1997), and the Spanish law on data protection [Regulation (EU) 2016/679 of the European Parliament and of the Council of April 27, 2016 on the protection of natural persons with regard to the processing of personal data and the free circulation of such data, and repealing Directive 95/46/EC (General Data Protection Regulation). All collected data received code numbers to maintain privacy and confidentiality.

### 4.3. Psychiatric and Neuropsychological Evaluation

The Spanish version of the PRISM (Psychiatric Research Interview for Substance and Mental Diseases) diagnostic interview was used for the evaluation of SUDs and other psychiatric disorders [81]. The neuropsychological battery was performed using different tests that have been demonstrated to be the most appropriate for detection of cognitive impairment in these patients: (1) Montreal Cognitive Assessment (MoCA) to assess of general cognitive status [82]; (2) Verbal Learning Test Spain-Complutense (VLTSC) to evaluate of verbal episodic memory [83]; (3) Trail-Making Test B [TMT B] to assess executive function [84]; (4) Rey–Osterrieth complex figure test (ROCF) to evaluate visual memory [85]; and (5) Digit span subtest (WAIS-IV) to evaluate short-term memory and verbal working memory [86]. We defined a value of −1 standard (Z) score as the cut-off point for mild cognitive impairment and a value of −2 standard score as the cut-off point for severe cognitive impairment as reported in previous studies [87].

All psychiatric and neuropsychological evaluations were performed in the morning. The entire clinical staff strictly adhered to a specified evaluation design so that the tests did not interfere with each other and to reduce the variability of testing between different clinicians. The tests were carried out in order: (1) sociodemographic and drug screening (PRISM), (2) MoCA, (3) VLTSC (first half), (4) ROCF (copy), (5) TMT B, (6) Digits, (7) ROCF (memory), and (8) VLTSC (second half). The session time was estimated at approximately 120 min.

### 4.4. Obtaining Plasma Samples

Blood samples were obtained in the morning after an 8–12 h fast (before psychiatric interviews). Venous blood was extracted into 10 mL K2 EDTA tubes (BD, Franklin Lakes, NJ, USA) and immediately processed to obtain plasma. Blood samples were centrifuged rum at 2200× *g* for 15 min (4 °C). Finally, plasma samples were stored at −80 °C until further analysis.

### 4.5. NfL and BDNF Quantification

Light Chain Neurofilament (NfL) concentrations were determined using a digital enzyme immunoassay and the SIMOA HD1 Analyzer platform [88]. Plasma BDNF concentrations were measured using a human custom 7-ProcartaPlex bead immunoassay kit (Invitrogen, cat. no. PPX-07-MXH6ANW) on a Luminex xMAP^®^ technology—MAGPIG system (ThermoFisher, Waltham, MA, USA). Sensitivity was approximately 57 pg/mL; mean intra-assay variation (%CV replicates) was 12.1%. The value of minimum detectable concentration was attributed to missing values below the standard curve [29].

### 4.6. Control of Possible Confounding Factors in the Analysis of NfL, BDNF and NFL/BDNF Ratio

#### 4.6.1. Influence of Age in the SUD, Dementia and Control Groups

Spearman correlations (rho) were performed to confirm previous literature about changes in NFL levels throughout life. When we analyzed NfL and age in the total sample, we observed a positive and significant correlation (rho = 0.700, *p* < 0.001), as other authors have also indicated [40]. Thus, we introduced this variable as a covariate in the statistical analysis.

#### 4.6.2. Influence of Abstinence Duration in the SUD Group

Spearman correlations (rho) were performed to eliminate the possibility that the lenght of abstinence from any drug of abuse could affect plasma concentrations of NfL and BDNF [63]. Our results showed that the mean duration of abstinence from all abused substances were high (>1 month). Moreover, we did not find any significant correlation between the length of abstinence of any abused substance and NfL or BDNF levels. Correlation analysis between plasma concentrations of NfL and BDNF according to addiction-related variables in the SUD group can be found in Appendix A.

#### 4.6.3. Influence of Comorbid Psychiatric Disorders and Psychotropic Medication in the SUD Group

We examined the effect of comorbid psychiatric disorders and psychotropic medication with the aim of excluding any effect that these variables might exert over plasma concentrations of NfL and BDNF [45,46,47]. We used a one-way ANCOVA with “group” (i.e., with comorbid psychiatric disorder and without comorbid psychiatric disorder) as a factor and age as a covariate. We did not observe significant differences in NfL and BDNF levels in comorbid psychiatric disorders or use of psychotropic medication in SUD patients. However, we investigated if specific comorbid mental disorders (mood and anxiety disorders) or psychiatric medication (antidepressants and anxiolytics), highly prevalent in the SUD group (see Table 2), could affect concentrations of these biological parameters. NfL levels were significantly affected by “comorbid anxiety disorder” factor (F_1,57_ = 4.895, *p* < 0.031, ηp^2^ = 0.079). Plasma concentrations of NfL were augmented in SUD patients with comorbid anxiety disorder compared to those without comorbid anxiety disorder (*p* < 0.031). Thus, we introduced this variable as a covariate in the analysis of NfL concentrations and cognitive impairment in SUD patients. However, comorbid psychiatric disorders and psychotropic medication did not affect BDNF levels or the NfL/BDNF ratio in the SUD group. Analysis between plasma concentrations of NfL and BDNF according to comorbid psychiatric disorders and psychotropic medication in the SUD group can be found in Appendix A.

### 4.7. Statistical Analysis

Significance of differences in qualitative variables was determined by Fisher’s exact test (Chi-square) and Mann–Whitney U test, respectively. Multiple analysis of covariance (ANCOVA) was performed to indicate the relative effects of explanatory variables (i.e., lifetime of SUDs, cognitive impairment) on plasma concentrations of NfL and BDNF controlling for age and comorbid anxiety disorder (when it was appropriated). Post hoc tests for multiple comparisons were performed using the Bonferroni correction test. Effect size was measured using partial eta squared (ηp^2^). Correlation analyses were performed using the Spearman’s coefficient (rho). The normal distribution of the variables was assessed using Lilliefors corrected Kolmogorov–Smirnov test. As the NfL, BDNF, and BDNF/NfL ratio variables in the study did not meet the assumption of normality assumptions (except for comorbid anxiety disorder analysis in the SUD group), logarithmic transformations (10) were used to preserve parametric assumptions. Then, the antilogarithm of the concentrations was used to represent them in the figures. Binary logistic regression analysis was performed using Pearson’s Chi-square (χ^2^) test with the Hosmer–Lemeshow test. Multicollinearity was assessed by examining Tolerance and Variance Inflation Factor (VIF). The cut-off value for Tolerance was >0.10 and <10 for VIF. Statistical analyses were carried out using GraphPad Prism version 5.04 and IBM SPSS Statistic version 22 (IBM, Armonk, NY, USA). A *p* value < 0.05 was considered statistically significant.

## Figures and Tables

**Figure 1 ijms-24-01183-f001:**
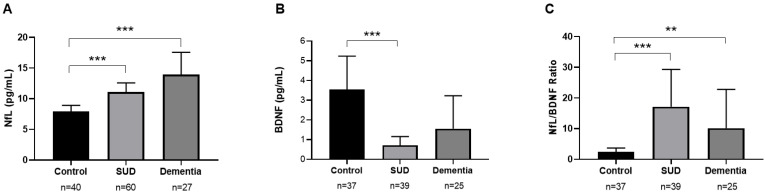
Plasma concentrations of neurofilament light chain ((**A**), NfL), Brain-derived neurotrophic factor ((**B**), BDNF) and ratio of NfL/BDNF (**C**) according to experimental groups. Data were analyzed by one-way analysis of covariance (ANCOVA). Bars are estimated marginal means and 95% confidence intervals. *** *p* < 0.001 and ** *p* < 0.010 denote a significant main effect of the group factor.

**Figure 2 ijms-24-01183-f002:**
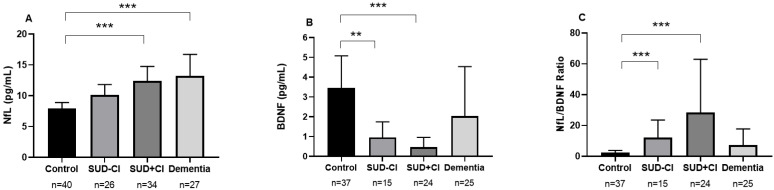
Plasma concentrations of neurofilament light chain ((**A**), NfL), Brain-derived neurotrophic factor ((**B**), BDNF) and ratio of NfL/BDNF (**C**) according to SUD + CI, SUD − CI, dementia and control groups. Data were analyzed by one-way analysis of covariance (ANCOVA). Bars are estimated marginal means and 95% confidence intervals. *** *p* < 0.001 and ** *p* < 0.010 denote a significant main effect of the group factor.

**Figure 3 ijms-24-01183-f003:**
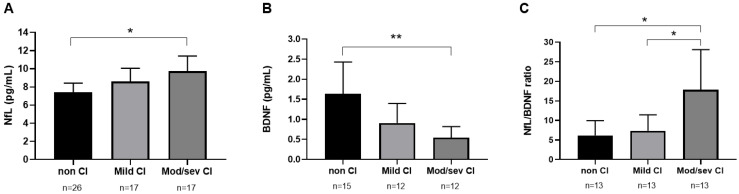
Plasma concentrations of neurofilament light chain ((**A**), NfL), Brain-derived neurotrophic factor ((**B**), BDNF) and ratio of NfL/BDNF (**C**) according to cognitive impairment in SUD patients. Data were analyzed by one-way analysis of covariance (ANCOVA). Bars are estimated marginal means and 95% confidence intervals representing. * *p* < 0.05 and ** *p* < 0.01 denote a significant main effect of the cognitive impairment factor. CI = cognitive impairment.

**Figure 4 ijms-24-01183-f004:**
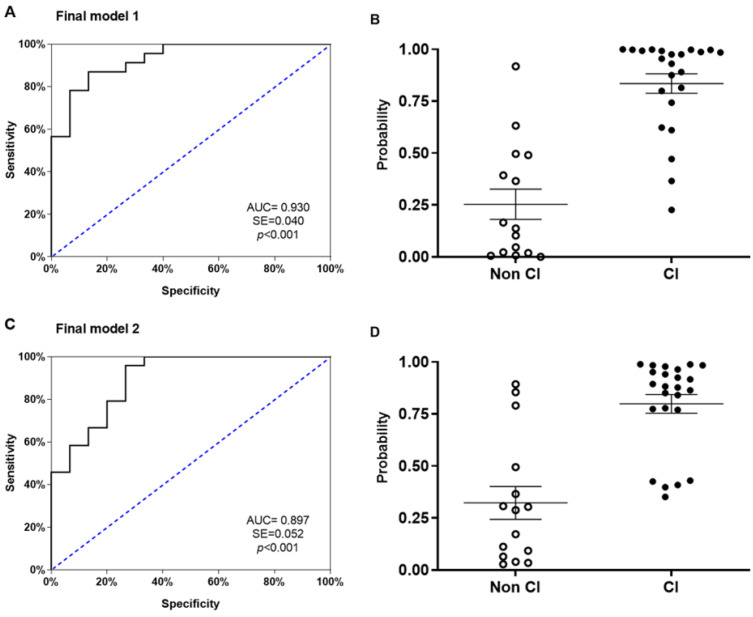
ROC analyses and scatter dots for multivariate predictive of final models of cognitive impairment. (**A**) ROC curve for the final model 1 whose variables were: “age”, “NfL/BDNF ratio”, “age at first alcohol use”, “age at onset of AUD”, and “length of AUD diagnosis”. (**B**) Scatter plot of the predictive probabilities for the final model 1. (**C**) ROC curve for the final model 2 whose variables were: “age” and “NfL/BDNF ratio”. (**D**) Scatter plot of the predictive probabilities for the final model 2. The lines of the scatter plots are means and standard deviations. CI = cognitive impairment. Open circles are patients without cognitive impairment (Non CI), black filled circles represent patients with cognitive impairment (CI).

**Table 1 ijms-24-01183-t001:** Socio-demographic characteristics of the sample.

Total Sample N = 127					
Variables		Control Group N = 40	SUD Group N = 60	Dementia Group N = 27	*p* Value
**Age** *(Mean* ± *SD)*	*Years*	52.25 ± 2.28	41.37 ± 12.40	75.33 ± 5.41	**<0.001 ^1^**
**Body Mass Index** *(Mean* ± *SD)*	*Kg/m^2^*	27.71 ± 4.05	26.47 ± 5.74	25.79 ± 3.15	0.320 ^2^
**Sex** *[N (%)]*	*Women*	20 (50)	10 (16.70)	12 (44.40)	**0.001 ^3^**
	*Men*	20 (50)	50 (83.30)	15 (55.60)	
**Education Level *** *[N (%)]*	*Low*	14 (35)	20 (35)	19 (70.40)	**0.002 ^3^**
	*Medium*	12 (30)	28 (46.70)	6 (22.20)	
	*High*	13 (32.50)	11 (18.30)	1 (3.70)	
**Occupation *** *[N (%)]*	*Employed*	33 (82.50)	17 (28.30)	1 (3.70)	**<0.001 ^3^**
	*Unemployed*	3 (7.50)	27 (45)	-	
	*Retired*	-	11 (18.30)	24 (88.9)	
	*Sick leave*	-	5 (8.30)	-	
	*Housework*	4 (10)	-	1 (3.70)	

^1^ Value was calculated with Kruskal–Wallis’s test. ^2^ Value was calculated with ANOVA’s *t* test. ^3^ Value was calculated with Fisher’s exact test or chi-squared test. Bold values are statistically significant when *p* < 0.05. (*) Some data were missing in the educational level and occupation variables.

**Table 2 ijms-24-01183-t002:** Clinical characteristics of the SUD group.

Variables		SUD Group N = 60
**Substance use disorders** *[N (%)]*	*Alcohol*	47 (78.30)
	*Cocaine*	36 (60)
	*Cannabis*	25 (41.70)
	*Sedatives*	8 (13.30)
	*Opioids*	7 (11.70)
	*AUD + other SUD*	33 (70.20)
**Comorbid psychiatric disorders** *N (%)]*	*Mood*	31 (51.70)
	*Anxiety*	21 (35)
	*Personality*	12 (20)
	*Psychotic*	5 (8.30)
**Psychiatric medication** *[N (%)]*	*Antidepressants*	26 (43.30)
	*Anxiolytics*	34 (56.70)
	*Antipsychotics*	9 (15)
	*Anticraving*	5 (8.3)
	*Disulfiram*	11 (18.30)
**Cognitive impairment (MoCA)** *[N (%)]*	*No*	26 (43.30)
	*Mild*	16 (26.70)
	*Moderate*	13 (21.70)
	*Severe*	5 (8.30)

**Table 3 ijms-24-01183-t003:** Neuropsychological psychometric values and their correlations with plasma concentrations of NfL and BDNF in SUD patients.

**Variables**		**SUD Group with Cognitive Impairment** **N = 34**			
		**Psychometric Data**		**Correlation Analyses**	
		**Direct Score** **[Mean (SD)]**	**Z Score** **[Mean (SD)]**	**NfL** **[Rho (*p* Value)]**	**BDNF** **[Rho (*p* Value)]**
**VLTSC**	*Immediate free recall (list A)*	37.09	−1.30 *	−0.279 (0.166)	0.058 (0.792)
	*Immediate free recall (list B)*	4.42	−1.09 *	**−0.363 (0.038)**	0.271 (0.211)
	*Short delay free recall*	8.42	−1.39 *	−0.316 (0.073)	0.347 (0.104)
	*Short delay cued recall*	9.88	−1.30 *	0.225 (0.209)	0.312 (0.147)
	*Long delay free recall*	10.06	−1.03 *	0.000 (0.999)	0.197 (0.367)
	*Long delay cued recall*	10.36	−1.18 *	0.095 (0.598)	0.049 (0.825)
	*Semantic strategies in immediate recall*	8.91	−0.76	−0.251 (0.159)	0.208 (0.342)
	*Semantic strategies in short delay free recall*	2.36	−0.76	−0.135 (0.455)	0.010 (0.965)
	*Semantic strategies in long delay free recall*	3.27	−0.70	−0.880 (0.626)	−0.038 (0.863)
	*Serial strategies in immediate recall*	2.94	−0.45	**−0.374 (0.032)**	0.215 (0.325)
	*Serial strategies in short delay free recall*	0.30	−0.42	**−0.464 (0.007)**	0.143 (0.516)
	*Serial strategies in long delay free recall*	0.15	−0.55	−0.190 (0.290)	0.000 (1)
	*Recognition*	14.30	−0.33	−0.158 (0.380)	0.293 (0.175)
	*Intrusions in free recall*	5.48	0.61	**0.408 (0.018)**	−0.083 (0.707)
	*Intrusions in cued recall*	3.12	0.73	**0.362 (0.038)**	−0.214 (0.327)
	*Perseverations*	3.58	−0.42	−0.266 (0.135)	0.126 (0.565)
** *ROCF* **	*Time (minutes)*	2.97	−0.154	0.209 (0.297)	−0.206 (0.383)
	*Figure*	33.80	0.96	−0.297 (0.124)	−0.058 (0.807)
	*Memory*	19.35	−0.10	−0.306 (0.121)	0.402 (0.088)
** *TMT B* **	*Time (seconds)*	106.26	3.37 **	0.349 (0.074)	−0.315 (0.190)
** *DIGITS SPAN* **	*Direct digits span*	4.89	−1.18 *	−0.131 (0.516)	−0.218 (0.371)
	*Backward digits span*	4.15	−0.49	−0.202 (0.311)	−0.309 (0.198)

(*) Values show Z-scores below −1 as cut-off point for mild cognitive impairment. (**) Value shows Z-score below −2 as cut-off point for severe cognitive impairment. Bold values are statistically significant for *p* < 0.05.

## Data Availability

Not applicable.

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
