# Peer review of "Plasma Concentrations of Neurofilament Light Chain Protein and Brain-Derived Neurotrophic Factor as Consistent Biomarkers of Cognitive Impairment in Alcohol Use Disorder"

_ijms, 2023, doi:10.3390/ijms24021183_

Round 1
Reviewer 1 Report
General Comments and Suggestions
This is a good contribution showing that some of the biomarkers for dementia from SUD and dementia from AD are the same. This thrust of the manuscript is important for several reasons. For one, it suggests that the cognitive decline in SUD and in AD may have its underlying origins in overlapping cellular mechanisms. As such, it also opens the door for more investigations of this kind on yet other types of dementia, potentially help to identify parts of the underlying cellular and molecular processes of all types of dementia.
In that light, the weakest link is the inclusion of cocaine and cannabinoids. First, there was no description on the recruitment of these patients. Second, one should always consider when to leave well enough alone. There is little advantage to include these patients but problems. I therefore strongly suggest that the authors delete all mentions of this group of patients in the present manuscript. Instead, consider another stand-alone project to explore whether in these patients the signatures of biomarkers would suggest the absence of dementia.
Otherwise, I like the general direction of this manuscript and the way the investigation is carried out. I suggest acceptance with moderate revisions to remove mentions of patients that abuse cocaine and cannabinoids.
Other Comments and Suggestions
1. People remember the early controversial studies of the effect of drinking leading to cerebellar degeneration. That was settled when the specific role of thiamine was discovered. For that reason, thiamine should be relevant here. In all of the manuscript, thiamine deficiency was mentioned only once in the Introduction and without any references. I suggest that the authors dig in the role of thiamine on SUD related dementia as well.
2. Also in the Introduction, BDNF was introduced abruptly and without linking to NfL. I wish the writing could take care of this.
3. Double check on matching. For example, in the first sentence under Discussion, “concentrations” and “is.”
Author Response
REFEREE 1
Response: We are very grateful for the review of our article. The suggestions have helped us to improve the deficiencies detected in our article. Therefore, we have made several changes throughout the manuscript in order to improve its quality. We have not underlined the changes in yellow because we have modified almost the entire article (introduction, method, results and discussion).
General Comments and Suggestions
This is a good contribution showing that some of the biomarkers for dementia from SUD and dementia from AD are the same. This thrust of the manuscript is important for several reasons. For one, it suggests that the cognitive decline in SUD and in AD may have its underlying origins in overlapping cellular mechanisms. As such, it also opens the door for more investigations of this kind on yet other types of dementia, potentially help to identify parts of the underlying cellular and molecular processes of all types of dementia.
In that light, the weakest link is the inclusion of cocaine and cannabinoids. First, there was no description on the recruitment of these patients. Second, one should always consider when to leave well enough alone. There is little advantage to include these patients but problems. I therefore strongly suggest that the authors delete all mentions of this group of patients in the present manuscript. Instead, consider another stand-alone project to explore whether in these patients the signatures of biomarkers would suggest the absence of dementia.
Otherwise, I like the general direction of this manuscript and the way the investigation is carried out. I suggest acceptance with moderate revisions to remove mentions of patients that abuse cocaine and cannabinoids.
Response: We really appreciate your comment. We have removed the section in which NfL and BDNF were analysed according to the type of substance abuse using an ANCOVA. Instead, we have performed correlations of addiction-related variables (for alcohol, cocaine, cannabis…) with both parameters that are in the Supplementary Table 1. On the other hand, the description on the recruitment of these patients can be found in the methodology. In the recruitment section appears that the SUD group had to meet the following inclusion criteria: people ≥18 years in the abstinence phase. The concept of “substance use disorders” involves alcohol, cocaine, cannabis, sedatives and opioids use disorders. Moreover, the method shows that the SUD group were diagnosed by PRISM interview for the evaluation of SUDs and other psychiatric disorders. Nevertheless, as we understand all the limitations that the inclusion of these analysis triggers: 1) we have included cannabis and cocaine articles in the introduction; 2) we have replaced ANCOVA analysis by correlation analysis, and 3) we have acknowledged this problem as a limitation of this work.
Other Comments and Suggestions:
- People remember the early controversial studies of the effect of drinking leading to cerebellar degeneration. That was settled when the specific role of thiamine was discovered. For that reason, thiamine should be relevant here. In all of the manuscript, thiamine deficiency was mentioned only once in the Introduction and without any references. I suggest that the authors dig in the role of thiamine on SUD related dementia as well.
Response: We are thankful for this recommendation. We have included in the introduction references to scientific literature regarding the relationship in between alcohol-related dementia, thiamine and other nutritional deficiencies. Moreover, we have incorporated other etiologies of alcohol-induced neurocognitive disorder such as the presence of comorbidity with affective disorders, neuroinflammation and oxidative stress.
- Also in the Introduction, BDNF was introduced abruptly and without linking to NfL. I wish the writing could take care of this.
Response: We are thankful for this suggestion. We have modified the introduction to better connect BDNF with respect to current knowledge on plasma cicultating NfL as biomarkers of nurodegeneration .
- Double check on matching. For example, in the first sentence under Discussion, “concentrations” and “is.”
Response: we have changed this sentence and checked the manuscript for grammar errors.

Reviewer 2 Report
The authors propose an interesting cross-sectional study of plasmatic biomarkers comparing long-term abstinent patients with a past history of SUD and patients with various neurodegenerative disorders. The comparison is performed on one selected neurodegenerative disorders biomarker (NfL) and one broad candidate biomarker of several psychiatric conditions (BDNF).
The method (a refined patented immuno-assay by Quanterix (SIMOA)) is very sensitive and allows to assess in the plasma very small amounts of proteins that were previously only detectable in the CSF. The methodology used for this paper is up to date, and the results are interesting data in a fast-moving field.
Nevertheless, several minor and major points should be improved or, if they cannot be improved, should be acknowledged as limitations of this work.
INTRODUCTION
The only level of alcohol use associated with 0 risk is 0. The u-shape or j-shape cognitive risk curves are obtained in prospective general population studies (5-7) with so demanding design that subjects with low socio-economic status in general and more specifically subjects with AUD are excluded. The only prospective study that was conducted not in a selected cohort of healthy subjects derived from the general population but from medical records in a country with universal care access (cited as REF #4 by the authors) clearly identifies alcohol use disorder as the leading risk factors for all dementia causes, and especially for early onset dementias.
The mortality and brain morbidity associated with excessive alcohol use are clearly documented, see: · Guérin, S.; Alcohol-attributable mortality in France. Eur. J. Public Health 2013, 23, 588–593. [Google Scholar] [CrossRef] [PubMed]
· Shield, K.; et al. National, regional, and global burdens of disease from 2000 to 2016 attributable to alcohol use: A comparative risk assessment study. Lancet Public Health 2020, 5, e51–e61. [Google Scholar] [CrossRef]
The sentence “However, alcohol consumption is not considered an essential component in the etiology of dementia except for thiamine deficiency” should be rephrased. Maybe our colleagues do not have it in mind, but the evidence is clearly that AUD is the leading risk factor of all dementias (see REF#4).
The sentence: “The main criticism of these studies is the attribution of cognitive deficits to concomitant medical and psychiatric comorbidities in patients with
alcohol consumption throughout life as mood disorders or cardiovascular diseases [9]” should be rephrased. The study (REF #4) took into account other known risk factors for dementia such as cardiovascular diseases.
It is true that in front of a given patient with AUD and persistent cognitive impairment, identifying the etiology is a complex inquiry (see : Azuar et al.. Alcohol Clin Exp Res. 2021 Mar;45(3):561-565. doi: 10.1111/acer.14554.. PMID: 33486797), and a given patient may have at the same time AUD, a neurodegenerative disorder, and several other risk factors for cognitive impairment that are associated with AUD.
Among them, apart from thiamine, the authors should cite other nutritional factors highly associated with severe AUD (see: Gautron MA et al. PMID: 29136089 ; Clergue-Duval et al. PMID: 34942994). Patients with severe AUD also display a lot of complications of their AUD with a cognitive impact, including the neurotoxic effect of chronic heavy alcohol use, cardio-vascular risk factors, stroke, epilepsy, the neurotoxic effect of episodes of alcohol withdrawal itself (see a recent paper, not yet appearing on PubMed: Clergue-Duval et al 202: https://www.mdpi.com/2076-3921/11/10/2078/htm; and not only thiamine deficiency.
“On the other hand, the background on cognitive impairment and dementia-related processes in cocaine and cannabis use disorder is still scarce and inconsistent [10,11].”
My comment on this sentence would be: the cognitive load of cocaine and cannabis use is also expected to be much lower knowing the heavy cognitive risk associated with AUD (see: above).
The authors cite their previous work on BDNF as a biomarker of AUD and cognitive impairment, but they should also mention straight from the introduction that plasmatic BDNF has been proposed as a broad biomarker of several psychiatric disorders that are highly comorbid with AUD and/or cognitive impairment, especially: depression (low plasma level during the episodes, increased in remission? See PMID: 36186501 ; elevated in various anxiety disorders such as PTSD see PMID: 33152026; and also evoked as a biomarker of cocaine use disorder see PMID: 25734326; . PMID: 24331739).
Regarding NFL, the authors would certainly be interested to cite a recent publication not yet on PUBMED but that can be downloaded from the publisher website, showing an association with the severity of alcohol withdrawal symptoms in patients with AUD: clergue-duval et al. 2022 https://onlinelibrary.wiley.com/doi/10.1111/adb.13232?utm_source=google&utm_medium=paidsearch&utm_campaign=R3MR425&utm_content=Medicine
RESULTS:
The table provided in the supplementary material contains very important clinical data regarding the group of patients with SUD and should be available in the main documents.
Especially, some of those variables deserve a specific attention as they may constitute confusion factors favoring the elevation of NFL or decrease in BDNF.
METHODS:
It is not clear by reading the method section I cannot find the information on when were the psychiatric and neuropsychological tests performed in AUD.
Knowing that they may be not only a cumulative effect of chronic alcohol use on cognition, but an effect of alcohol withdrawal episodes themselves, it would be important i) to know if the time of the neuropsychological assessment was standardized, if the data is not available, please state so as a limit; ii) if all patients were assessed in abstinence of all substances; iii) to test the correlation between NFL and BDNF and the length of abstinence (more than 1 year for alcohol as I read in the supplementary table).
I would have like to read a direct comparison of patients with SUD without CI and controls and dementia.
The severity of SUD patients with cognitive impairment is hard to find for the reader : the MoCA score should figure in a line of the table describing the patients.
The effect of depression or antidepressant treatment on the BDNF score and thus on the NFL/BDNF ratio should be tested in the regression as a possible confounding factor.
Discussion: the possible confounding roles of time from abstinence and depression should be discussed with regard to the previously suggested literature.
The authors should be more cautious while discussing their results as they were obtained in a cross sectional design. The need for prospective assessment whould be emphasize.
For exemple, the sentence “Thus, we show that plasma concentrations of NfL increase according to the duration of AUD throughout life” is not true.
This would require a prospective design. The data only show that in this sample, patients with longer history of AUD have higher NfL levels.
“The compensatory BDNF mechanism declines with progressively increasing alcohol severity” is not shown by this design. It could be due to other confounding factors. Please rephrase.
“Finally, plasma concentrations of NfL and BDNF and certain alcohol-related variables proved to be robust factors to detect cognitive decline”: the results do not show that.
Overall, an interesting cross-sectional study in a fast-moving field.
The authors should perform some further statistical analysis to control for well known confounding factors. They should interpret their result with more caution while comparing them to the recently available data.

Author Response
ANSWER TO REFEREE 2
Response: We are very grateful for the extensive revision made on the report. All suggestions have been taking in consideration, and they have indeed helped us to improve it. Since the changes made throughout the manuscript were very extensive, we did not underline them in yellow to improve reading. What we have done is to include in this revision the whole paragraphs modified that belongs to the introduction, methods, results and discussion sections.
INTRODUCTION
1-The only level of alcohol use associated with 0 risk is 0. The u-shape or j-shape cognitive risk curves are obtained in prospective general population studies (5-7) with so demanding design that subjects with low socio-economic status in general and more specifically subjects with AUD are excluded. The only prospective study that was conducted not in a selected cohort of healthy subjects derived from the general population but from medical records in a country with universal care access (cited as REF #4 by the authors) clearly identifies alcohol use disorder as the leading risk factors for all dementia causes, and especially for early onset dementias.
The mortality and brain morbidity associated with excessive alcohol use are clearly documented, see: · Guérin, S.; Alcohol-attributable mortality in France. Eur. J. Public Health 2013, 23, 588–593. [Google Scholar] [CrossRef] [PubMed]
- Shield, K.; et al. National, regional, and global burdens of disease from 2000 to 2016 attributable to alcohol use: A comparative risk assessment study. Lancet Public Health 2020, 5, e51–e61. [Google Scholar] [CrossRef]
Response: We welcome these recommendations. As suggested we modified the introduction, removing the references that might lead to the misperception about the risk of alcohol use on cognitive processes. In addition, we have included in the introduction the suggested articles on brain morbidity and/or mortality.
2- The sentence “However, alcohol consumption is not considered an essential component in the etiology of dementia except for thiamine deficiency” should be rephrased. Maybe our colleagues do not have it in mind, but the evidence is clearly that AUD is the leading risk factor of all dementias (see REF#4).
Response: Following the above described line, we have removed this sentence to clearly support the prominent role of alcohol abuse in dementia.
The sentence: “The main criticism of these studies is the attribution of cognitive deficits to concomitant medical and psychiatric comorbidities in patients with alcohol consumption throughout life as mood disorders or cardiovascular diseases [9]” should be rephrased. The study (REF #4) took into account other known risk factors for dementia such as cardiovascular diseases.
Response: We agree with the referee, so we have modified this sentence accordingly.
3-It is true that in front of a given patient with AUD and persistent cognitive impairment, identifying the etiology is a complex inquiry (see : Azuar et al.. Alcohol Clin Exp Res. 2021 Mar;45(3):561-565. doi: 10.1111/acer.14554.. PMID: 33486797), and a given patient may have at the same time AUD, a neurodegenerative disorder, and several other risk factors for cognitive impairment that are associated
Among them, apart from thiamine, the authors should cite other nutritional factors highly associated with severe AUD (see: Gautron MA et al. PMID: 29136089 ; Clergue-Duval et al. PMID: 34942994). Patients with severe AUD also display a lot of complications of their AUD with a cognitive impact, including the neurotoxic effect of chronic heavy alcohol use, cardio-vascular risk factors, stroke, epilepsy, the neurotoxic effect of episodes of alcohol withdrawal itself (see a recent paper, not yet appearing on PubMed: Clergue-Duval et al 202: https://www.mdpi.com/2076-3921/11/10/2078/htm; and not only.
Response: We really appreciate the suggestions of the referee. All these contributing factors are now enumerated and the suggested references cited in the introduction. Thus, we now explain that substance-induce neurocognitive disorder may have different etiologies such as other nutritional factors, comorbidity with affective disorders, effect of early abstinence, neuroinflammation and oxidative stress.
4-“On the other hand, the background on cognitive impairment and dementia-related processes in cocaine and cannabis use disorder is still scarce and inconsistent [10,11].” My comment on this sentence would be: the cognitive load of cocaine and cannabis use is also expected to be much lower knowing the heavy cognitive risk associated with AUD (see: above).
Response: We have not removed this sentence. Instead, we have included scientific literature about cocaine and cannabis cognitive impairment in the introduction:
“On the other hand, the background on cognitive impairment and dementia-related processes in cocaine and cannabis use disorders is still scarce and inconsistent [30-31]. Although some studies suggest that prolonged cocaine use is related with deterioration in some cognitive domains, these could be attributable to concomitant alcohol consumption [32]. However, patients with cocaine abuse manifest severe cocaine-induced cardiovascular consequences such as vasoconstriction, endothelial dysfunction, arteriosclerosis [33] and increases in oxidative stress [34] that could underlie cognitive dysfunction [35,36]. Moreover, acute cocaine use could also be a contributing factor to stroke risk in young people [37]. Similarly, although heavy cannabis use appears to lead to neuropsychological impairment, these associations might be attenuated or not significant when confounding variables are controlled (i.e., other substance use, psychiatric disorders or psychosocial variables) [38]”.
5- The authors cite their previous work on BDNF as a biomarker of AUD and cognitive impairment, but they should also mention straight from the introduction that plasmatic BDNF has been proposed as a broad biomarker of several psychiatric disorders that are highly comorbid with AUD and/or cognitive impairment, especially: depression (low plasma level during the episodes, increased in remission? See Claro AE, PMID: 36186501 ; elevated in various anxiety disorders such as PTSD see Mojtabavi H, et al. PMID: 33152026; and also evoked as a biomarker of cocaine use disorder see Pedraz M, et al. PMID: 25734326; Viola TW, et al. . PMID: 24331739).
Response: We are glad to include this modification in the article. We have introduced your suggested articles (among others) in the introduction:
“However, BDNF has been proposed as a broad biomarker of several neuropsychiatric disorders that are highly comorbid with AUD and/or cognitive impairment, especially depression [45], posttraumatic stress disorder [46], schizophrenia [47], cocaine use disorder [48,49], Parkinson’s disease [50] and Alzheimer’s disease [51].”
5-Regarding NFL, the authors would certainly be interested to cite a recent publication not yet on PUBMED but that can be downloaded from the publisher website, showing an association with the severity of alcohol withdrawal symptoms in patients with AUD: clergue-duval et al. 2022 https://onlinelibrary.wiley.com/doi/10.1111/adb.13232?utm_source=google&utm_medium=paidsearch&utm_campaign=R3MR425&utm_content=Medicine
Response: We have introduced your suggested article with another new article about NfL and alcohol in the introduction and the discussion. Li, Y., Duan, R., Gong, Z., Jing, L., Zhang, T., Zhang, Y., & Jia, Y. (2021). Neurofilament Light Chain Is a Promising Biomarker in Alcohol Dependence. Frontiers in psychiatry, 12, 754969. https://doi.org/10.3389/fpsyt.2021.754969
RESULTS:
6-The table provided in the supplementary material contains very important clinical data regarding the group of patients with SUD and should be available in the main documents.
Response: We have moved this table into the corresponding results section.
7-Especially, some of those variables deserve a specific attention as they may constitute confusion factors favoring the elevation of NFL or decrease in BDNF.
Response: We are thankful for this great suggestion. We have made a new table with descriptive statistics about addiction-related variables and the main correlations with NfL and BDNF. This table is in Supplementary Table 1 and its main results are described in the corresponding section:
“Correlation analysis between addiction-related variables and plasma concentrations of NfL and BDNF was performed to explore the effect of specific substance of abuse on these parameters using partial correlations controlling for age (rho) (Table 3). Interestingly, we found significant correlations between NfL concentrations and age at first alcohol use (rho=0.315, p=0.031), age at onset of AUD (rho= 0.455, p=0.001) and length of AUD diagnosis (rho=0.375, p=0.010). We also showed significant correlations between NfL concentrations and age at first cocaine use (rho=0.437, p=0.008) and with the severity of sedative use disorder (rho=-0.815, p=0.014). In addition, we find a positive and significant correlation between BDNF levels and the severity of cannabis use disorder (rho=0.605, p=0.017). Correlation analysis between plasma concentrations of NfL and BDNF according to addiction-related variables in the SUD group is in Supplementary Table 1”
METHODS:
8-It is not clear by reading the method section I cannot find the information on when were the psychiatric and neuropsychological tests performed in AUD.
Response: We thank the reviewer for the comment. We have clarify the protocol and the standardization of the psychiatric and neuropsychological evaluation. We have included this issue in the method section of the article.
“All psychiatric and neuropsychological evaluations were performed in the morning. The entire clinical staff strictly adhered to a specified evaluation design so that the tests did not interfere with each other and to reduce the variability of testing between different clinicians. The tests were carried out in order: 1) sociodemographic and drug screening (PRISM), 2) MoCA, 3) TAVEC (1st half), 4) ROCF (copy), 5) TMT B, 6) Digits, 7) ROCF (memory), 8) TAVEC (2nd half). The session time is estimated at approximately 120 minutes.”
9-Knowing that they may be not only a cumulative effect of chronic alcohol use on cognition, but an effect of alcohol withdrawal episodes themselves, it would be important i) to know if the time of the neuropsychological assessment was standardized, if the data is not available, please state so as a limit; ii) if all patients were assessed in abstinence of all substances; iii) to test the correlation between NFL and BDNF and the length of abstinence (more than 1 year for alcohol as I read in the supplementary table).
Response: We thank the reviewer for such an important suggestion. First, as we mentioned above, all psychiatric and neuropsychological evaluations were performed in the morning with the same protocol in each patient, so this variable can be under control. Second, as shown in the new Supplementary Table 2, the abstinence mean of all substances was quite high for all substances (>1 month). Moreover, to confirm if the abstinence of some drug of abuse was interfering with the effect of SUD on NfL or BDNF levels we have performed an additional analysis (see below in the table). Thus, NfL and BDNF did not correlate with the abstinence of any substance.
Supplementary Table 1. Correlation analysis according to addiction-related variables and NfL plasma concentrations.
VARIABLES |
SUD group N=60 |
Correlation Analyses |
|||
NfL [Rho (p value)] |
BDNF [Rho (p value)] |
||||
Age at first drug use [Mean (SD)] |
Years |
Alcohol Cocaine Cannabis Sedatives Opioids |
14.04 (4.31) 19.22 (5.05) 14.60 (2.48) 18.50 (1.77) 20.86 (8.44) |
0.315 (0.031) 0.437 (0.008) 0.170 (0.416) 0.491 (0.217) 0.306 (0.504) |
-0.079 (0.651) -0.103 (0.640) -0.146 (0.603) 0.103 (0.870) 0.600 (0.285) |
|
|
|
|
|
|
Age at onset of SUD [Mean (SD)] |
Years |
Alcohol Cocaine Cannabis Sedatives Opioids |
17.85 (9.31) 24.75 (8.38) 16.92 (4.04) 24.25 (13.46) 22.14 (7.99) |
0.455 (0.001) 0.304 (0.071) 0.144 (0.493) 0.596 (0.119) 0.252 (0.585) |
-0.229 (0.189) -0.189 (0.387) -0.162 (0.565) -0.447 (0.450) 0.564 (0.322) |
|
|
|
|
|
|
Length of SUD diagnosis [Mean (SD)] |
Years |
Alcohol Cocaine Cannabis Sedatives Opioids |
16.70 (12.06) 10.93 (8.65) 13 (10.01) 7.88 (5.94) 14.57 (13.54) |
0.375 (0.010) 0.251 (0.140) 0.231 (0.266) 0.217 (0.606) 0.198 (0.670) |
-0.229 (0.192) -0.104 (0.638) -0.005 (0.985) -0.053 (0.933) -0.500 (0.391) |
|
|
|
|
|
|
Severity criteria [Mean (SD)] |
Criteria [1-11] |
Alcohol Cocaine Cannabis Sedatives Opioids |
6.35 (2.91) 6.11 (3.66) 5.73 (3) 7 (2.98) 4 (4.36) |
0.032 (0.831) -0.246 (0.148) -0.383 (0.059) -0.815 (0.014) -0.299 (0.514) |
0.106 (0.543) 0.087 (0.694) 0.605 (0.017) 0.359 (0.553) -0.100 (0.873) |
|
|
|
|
|
|
Length of abstinence [Mean (SD)] |
Days |
Alcohol Cocaine Cannabis Sedatives Opioids |
400.24 (995.31) 216.67 (672.13) 526.68 (1488.22) 72.43 (167.31) 1103.43 (1290.84) |
0.121 (0.422) 0.250 (0.141) 0.159 (0.447) 0.729 (0.063) 0.500 (0.253) |
-0.163 (0.358) 0.133 (0.546) -0.094 (0.739) -0.632 (0.368) -0.105 (0.866) |
10-I would have like to read a direct comparison of patients with SUD without CI and controls and dementia.
Response: We really appreciate this recommendation. We have included this analysis in a new result section.
“We wanted to investigate how cognitive integrity (assessed by MoCA) in SUD patients (SUD+CI vs SUD-CI) could affect plasma concentrations of NfL, BDNF, and the NfL/BDNF ratio using a one-way ANCOVA with "group" (SUD+CI, SUD-CI, dementia and control groups) as a factor and age as a covariate. When we analyzed the SUD+CI group, we found that plasma concentrations of NfL, BDNF and NfL/BDNF ratio were significantly different between groups (F3,122=12.499, p<0.001, ηp²=0.235; F2,96=11.821, p<0.001, ηp²=0.270; F3,96=16.154, p<0.001, ηp²=0.335, respectively). We observed higher NfL levels in the SUD+CI (p<0.001) and dementia group (p=0.001) compared to the control group (Figure 2A). We also found lower BDNF levels in the SUD+CI (p<0.001) and in the SUD-CI group (p=0.003) compared to the control group (Figure 2B). NfL/BDNF ratio was increased in the SUD+CI group (p<0.001) and in the SUD-CI (p=0.001) compared to the control group (Figure 2C)”.
11-The severity of SUD patients with cognitive impairment is hard to find for the reader : the MoCA score should figure in a line of the table describing the patients.
Response: We really appreciate this comment. We have included the MoCa score description in Table 2.
Table 2. Clinical characteristics of the SUD group.
VARIABLES |
SUD group N=60 |
|
Substance use disorders [N (%)] |
Alcohol Cocaine Cannabis Sedatives Opioids AUD + other SUD |
47 (78.30) 36 (60) 25 (41.70) 8 (13.30) 7 (11.70) 33 (70.20) |
Comorbid psychiatric disorders N (%)] |
Mood Anxiety Personality Psychotic |
31 (51.70) 21 (35) 12 (20) 5 (8.30) |
Psychiatric medication [N (%)] |
Antidepressants Anxiolytics Antipsychotics Anticraving* Disulfiram |
26 (43.30) 34 (56.70) 9 (15) 5 (8.3) 11 (18.30) |
Cognitive impairment (MoCA) [N (%)] |
No Mild Moderate Severe |
26 (43.30) 16 (26.70) 13 (21.70) 5 (8.30) |
12-The effect of depression or antidepressant treatment on the BDNF score and thus on the NFL/BDNF ratio should be tested in the regression as a possible confounding factor.
Response: In the methodology, we have included several analyses to control if psychiatric disorders (depression/anxiety), psychotropic medication (antidepressants/anxiolytics) and age could affect NfL or BDNF concentrations. However, only anxiety disorders might affect NfL concentrations in the SUD group but not BDNF or NFL/BFNG ratio. Thus, we controlled for this variable in the analysis of the results section. Age of patients was also controlled. In addition, we ruled out the effect of abstinence on NfL or BDNF concentrations (see supplementary Table 1 above).
Control of Possible Confounding Factors in the Analysis of NfL, BDNF and NFL/BDNF Ratio
4.6.1. Influence of Age in the SUD, Dementia and Control groups.
Spearman correlations (rho) were performed to confirm previous literature about changes in NFL levels trought life. When we analized NfL and age in the total sample, we observed a positive and significant correlation (rho=0.700, p<0.001) as other authors have also indicated [40]. Thus, we introduced this variable as a covariate in the statistical analysis.
4.6.2. Influence of Abstinence Duration in the SUD Group
Spearman correlations (rho) were performed to eliminate the possibility that the lenght of abstinence from any drug of abuse could affect plasma concentrations of NfL and BDNF [63]. Our results showed that the mean duration of abstinence from all drug of abuse were high (>1 month). Moreover, we did not find any significant correlation between the length of abstinence of any substance of abuse and NfL or BDNF levels. Correlation analysis between plasma concentrations of NfL and BDNF according to addiction-related variables in the SUD group is in Supplementary Table 1.
4.6.3. Influence of Comorbid Psychiatric Disorders and Psichotropic Medication in the SUD Group
We examined the effect of comorbid psychiatric disorders and psichotropic medication with the aim of excluding any effect that these variables might exert over plasma concentratios of NfL and BDNF [45-47]. We used a one-way ANCOVA with “group” (i.e., with comorbid psychiatric disorder and without comorbid psychiatric disorder) as factor and age as covariate. We did not observe significant differences in NfL and BDNF levels in comorbid psychiatric disorders or use of psychotropic medication in SUD patients. However, we investigated if specific comorbid mental disorders (mood and anxiety disorders) or psychiatric medication (antidepressants and anxiolitics), highly prevalent in the SUD group (see Table 2), could affect concentrations of these biological parameters. NfL levels was significantly affected by “comorbid anxiety disorder” factor (F1,57=4.895, p<0.031, ηp²=0.079). Plasma concentrations of NfL were augmented in SUD patients with comorbid anxiety group compared to those without comorbid anxiety disorder (p<0.031). Thus, we introduced this variable as a covariate in the analysis of NfL concentrations and cognitive impairment in SUD patients. However, comorbid psychiatric disorders and psichotropic medication did not affect BDNF levels and NfL/BDNF ratio in the SUD group. Analysis between plasma concentrations of NfL and BDNF according to comorbid psychiatric disorders and psychotropic medication in the SUD group are in Supplementary Table 2, 3 and 4.
DISCUSSION
13-the possible confounding roles of time from abstinence and depression should be discussed with regard to the previously suggested literature.
Response: We have considered the confounding variables in the discussion.
“The present study supports that substance-induced neurocognitive disorder is not only associated with deficits in plastic/trophic factors (BDNF) but also with a real structural brain damage (NfL). It is important to note that we had to control for confounding variables that could alter plasma concentrations of NfL and BDNF such as psychiatric conditions (mood and anxiety disorders), psychotropic medication (antidepressants and anxiolytics), early withdrawal effects (days of abstinence) and age (years). Besides NfL evaluation can discriminate between depression from neurodegenerative disorders, our results indicated that anxiety disorders can affect plasma concentrations of NfL in our SUD sample [39]. Similarly, some studies have reported that depressive and anxiety symptoms could worsen NfL profile in neurodegenerative processes such as multiple sclerosis and Parkinson’s disease [55,56]. However, when we controlled anxiety disorders in the analysis, the association between cognitive impairment and NfL concentrations remains significant in SUD patients. Furthermore, we observed a positive and significant correlation between NfL and age of participants. NfL concentrations are not only a strong biomarker for neurodegeneration but are also closely related to senescence [40]. Unless for anxiety disorders and age, there were no other variables that affected levels of NfL, BDNF or NfL/BDNF ratio in our sample.”
14-The authors should be more cautious while discussing their results as they were obtained in a cross sectional design. The need for prospective assessment whould be emphasize.For example, the sentence “Thus, we show that plasma concentrations of NfL increase according to the duration of AUD throughout life” is not true. This would require a prospective design. The data only show that in this sample, patients with longer history of AUD have higher NfL levels.
Response: We are grateful to the reviewer for this recommendation. We have tried to be less presumptuous. Furthermore, we have considered the need for a prospective design in the limitation section.
“Nevertheless, this study has some limitations that should be considered in future research. First, a high sample size and prospective design are needed to test and establish NfL, BDNF and NfL/BDNF ratio as potential diagnostic biomarkers of neurodegeneration in SUD population. Secondly, a stratification of the SUD group is necessary according to the type of drug use (alcohol, cocaine, cannabis, etc.). Finally, the study lacks a significant representation of the female population, which precludes investigation of gender/sex differences in NfL and BDNF markers.”
15-“The compensatory BDNF mechanism declines with progressively increasing alcohol severity” is not shown by this design. It could be due to other confounding factors. Please rephrase.
Response: We have modified the statement and discussed in accordance with more scientific literature about this topic.
“This might indicate that BDNF could be a biomarker of both substance abuse and substance-related cognitive impairment as our group has reported in previous studies [43,44]. Interestingly, we did not observe differences in BDNF concentrations in dementia patients compared to controls. Thus, these results could suggest that disrupted trophic signaling could underlie the neuronal death and cognitive dysfunction observed in our SUD patients. In contrast, the neuroprotection of BDNF might not be enough to counteract brain damage in patients with dementia.”
[…]
“Our results showed that NfL/BDNF ratio was higher in SUD and dementia patients compared to controls. Increased NfL/BDNF ratio were observed in both SUD patients with and without cognitive impairment, but in those with moderate/severe cognitive impairment this index was especially higher. In other words, substance abuse throughout life (particularly alcohol) is not only toxic but also anti-regenerative. These results might indicate that BDNF could be involved in a compensatory mechanism against neuronal damage but not in advanced stages when neurocognitive disability is established. In this regard, an emerging line supports the role of BDNF/TrkB signaling pathway as compensatory responses that delay symptoms in the early stage of Alzheimer’s disease, while in later stages they would not be sufficient to prevent neurodegeneration ([70–72]).”
16-“Finally, plasma concentrations of NfL and BDNF and certain alcohol-related variables proved to be robust factors to detect cognitive decline”: the results do not show that.
Response: We have removed this sentence and we have changed this analysis too. We have performed a logistic regression model with the NfL/BDNF ratio in order to control anxiety effects on NfL concentrations. The results indicated that NfL/BDNF ratio was a good parameter to stratify or SUD sample in cognitive impaired and non-cognitive impaired patients.
“We generated a binary logistic regression model to discriminate between SUD patients with and without cognitive impairment (assessed by MoCA). In the final model, the variables included in the first step were “age”, “NfL/BDNF ratio”, “age at first alcohol use”, “age at onset of AUD”, “length of AUD diagnosis”, “age at first cocaine use”, “sedatives severity criteria” and “cannabis severity criteria”. Model was prepared using the backward stepwise method and the predictive covariates were restricted to five, which were “age”, “NfL/BDNF ratio”, “age at first alcohol use”, “age at onset of AUD” and “length of AUD diagnosis”. The Hosmer-Lemeshow test indicated good calibration (X2=5.905; p=0.658) and was able to explain the variation of the dependent variable in 69.7% of the cases according to the Nagelkerke R2 method. It had a classification percentage of 86.8% showing a high sensitivity to classify SUD patients with cognitive impairment (87%) and without cognitive impairment (86.7%). ROC curve analysis (AUC=0.930) indicated a high discrimination power (Figure 4A). The scatter plot of the predictive probabilities for SUD patients indicated that the means were significantly different between both groups (U=24, p<0.001) (Figure 4B).
Binary logistic regression analysis was then used to assess the potential for the NfL/BDNF ratio alone to be a good predictor for discriminating between SUD patients with and without cognitive impairment (assessed by MoCA). The variables included in the first step were “age” and “NfL/BDNF ratio”. Model was performed using the backward stepwise method and the predictive covariates were restricted to both variables. The Hosmer-Lemeshow test indicated good calibration (X2=11.114; p=0.195) and was able to explain the variation of the dependent variable in 58.2% of the cases according to the Nagelkerke R2 method. It had a classification percentage of 79.5% showing a high sensitivity to classify SUD patients with cognitive impairment (79.2%) and without cognitive impairment (80%). ROC curve analysis (AUC=0.897) indicated a high discrimination power (Figure 4C). The scatter plot of the predictive probabilities for SUD patients indicated that the means were significantly different between both groups (U=37, p<0.001) (Figure 4D).”
[…]
In addition, we tested the NfL/BDNF ratio as a potential biomarker for substance-induced neurocognitive disorder in two binary logistic regression. In the first model, “age”, “NfL/BDNF ratio”, “age at first alcohol use”, “age at onset of AUD” and “length of AUD diagnosis” were variables able to stratify our SUD sample in cognitive impaired and non-cognitive impaired patients in a 93%. In the second model, “age” and “NfL/BDNF ratio” were variables sufficiently capable of classifying our SUD sample in cognitive impaired and non-cognitive impaired patients in an 89%.
